# An Analytical Solution for Investigating the Characteristics of Tidal Wave and Surge Propagation Associated with Non-Tropical and Tropical Cyclones in the Humen Estuary, Pearl River

**Zhuo Zhang** [1,2]**, Fei Guo** [1,2,]*****, Di Hu** [1,2] **and Dong Zhang** [1,2]

1   Key Laboratory of Virtual Geographic Environment, Nanjing Normal University, Ministry of Education,
    Nanjing 210023, China; mercury1214@126.com (Z.Z.); 09374@njnu.edu.cn (D.H.);
    zhangdong@njnu.edu.cn (D.Z.)
2   Jiangsu Center for Collaborative Innovation in Geographical Information Resource Development
    and Application, Nanjing 210023, China
*   Correspondence: guofei@njnu.edu.cn

**Abstract:** The Humen Estuary, one of the largest outlets of the Pearl River, is a long and wide tidal channel with a considerable tidal flow every year. Storm surges, always superposing spring tide, travel from the estuary and endanger the safety of people living around the river. However, little research has quantified the relationship between the hydraulic characteristics and the geometry features in this estuary. In this regard, an analytical model, combined with a numerical model, is applied to investigate the characteristics of tidal waves and surge propagations in the estuary. Given the geometric, topographic, and tidal parameters at the mouth of the estuary, the tidal damping and wave celerity can be computed. The numerical results were used to calibrate and verify the analytical model. The results indicate that the analytical model can describe the astronomical tidal dynamics very well in correspondence with the numerical results. However, the analytical model cannot predict the tide well when a tropical cyclone-induced surge is superimposed on the astronomical tide. The reason is that this model does not take the wind stress and the pressure depression into account. After reducing Manning's coefficient, we found that the analytical results could be close to the numerical results. Finally, we analyzed the characteristics of the tidal wave in the Humen Estuary using the analytical solution and its parameters.

**Keywords:** tidal damping; wave propagation; analytical model; surge; pearl river

## 1. Introduction

Tidal waves are a major hydrodynamic factor that influences the flooding, ecosystems, transport, and morphological evolution in estuaries. In contrast to tidal waves in deep oceans, which are driven by astronomical forces with a high degree of predictability, waves propagating along estuaries tend to adjust their amplitude, celerity, and shape during interactions with topography, friction, and other secondary factors such as wind stress and runoff. A tidal wave in a shallow estuary has the characteristics of amplification and damping, which alternate along the estuary. The tidal amplitude is influenced by two dominant processes: amplification due to convergent cross sections with a decrease in depth in the landward direction and damping due to bottom friction. If the former process dominates over the latter, the wave is amplified; otherwise, the wave is damped. The celerity of propagation is also influenced by the processes of tidal damping and amplification. If the tidal wave is damped, the celerity is decreased. Conversely, if the tidal wave is amplified, the celerity is increased [1]. Moreover, the tidal wave can be strongly distorted as it propagates into the estuarine system [2]. Therefore, tidal wave propagation

in an estuary is an interesting but difficult issue to solve because of the nonlinear interaction resulting from the complex estuarine boundary lines and variably shallow topography.

To understand the mechanism of tidal wave propagation along an estuary, vast efforts have been made to induce and solve the estuary hydrodynamic model. Generally, these works can be divided into two methods: analytical solutions and numerical simulations. With advances in computational technology, tidal wave propagation can be accurately simulated by numerical models [2–5]. Compared with analytical solutions, numerical models are better at providing high-resolution data under various controlling conditions, which can be comparable with the measurements in a real estuary. However, the numerical results should be further investigated and extracted through the analytical solution to obtain a deeper understanding of the effect by multiple controlling factors with the respect of tidal wave propagation. Specifically, to figure out the relationship between hydrodynamics and the geometry of the estuary, we need the analytical solution not only to fit the data but also to extract some important parameters such as velocity number, damping number, celerity number, and their relationships with the geometry parameter of the estuary. Thus, an analytical solution is still an irreplaceable instrument in analyzing the properties of tidal waves propagating in an estuary. Thus far, a range of analytical solutions based on 1D Saint–Venant equations have been derived by Hunt [6](1964), Ippen [7], and Prandle [8], which are all for a prismatic estuary and river. Meanwhile, the solutions for a more widely convergent estuary have been presented by Godin [9], Jay [10], Lanzoni and Seminara [11], and VanRijn [12]. Most of the above investigators linearized the equations by means of the perturbation method in an Eulerian framework. In contrast, Savenije [13] derived a relatively simple solution in a Lagrangian framework. Furthermore, Savenije [14] presented a fully explicit solution of the tidal equations by solving the set of four implicit equations derived from the Saint–Venant equations. With the advantage of considering the nonlinear effect of the bottom friction, Savenije's analytical solution is constantly improved and widely used in analyzing the wave propagation on typical alluvial estuaries [15,16].

The Humen Estuary, one of the major accesses for a tidal wave from the South China Sea entering the Pearl River network, has a significant effect on flooding and inundation in Guangzhou city, which is the capital and economic center of Guangdong Province. However, limited measured data in the Humen Estuary restrict the understanding of tidal wave propagation along the Humen Estuary in the upstream direction. Although some previous numerical simulations have been performed [17,18], their results are not direct and clear enough to reflect the characteristics of how a certain controlling parameter affects other parameters. Another question is whether Savenije's analytical solution can be applied to the situation under the interaction of tides and surges during typhoon landfall. If not, what are the changes in the hydrodynamic system and which parameters should be adjusted in the analytical solution?

With the questions mentioned above, the objectives of the study are as follows: 1. to compose and validate a method of analysis that combines a classical analytical solution and the numerical results and is able to correctly reflect the characteristics of tide propagation in the Humen Estuary; 2. to check and verify a way to adapt the method of analysis to the context of tidal wave and surge propagation associated with tropical cyclones; and 3. to extract insight into the hydrodynamic features and factors in the Humen Estuary. Thus, we apply Savenije's analytical solution to the Humen Estuary in conjunction with numerical simulation results to describe and analyze the characteristics of the tidal wave in the Humen Estuary, Pearl River. A series of tests are conducted to answer the above questions in this paper.

The sections are organized as follows. First, some background information about the Humen Estuary in the Pearl River network is introduced in Section 2, followed by a brief introduction of Savenije's analytical solution in Section 3. In Section 4, the numerical model is verified. In Section 5, calibration and verification for astronomic tides and storm surges are carried out. In Section 6, the discussion focuses on the characteristics of tidal waves in the Humen Estuary. Finally, conclusions are drawn in Section 7.

## 2. Study Areas

### 2.1. Overview

The Pearl River network, located in South China, delivers a large amount of fresh water (ranging from 20,000 m$^3$/s in the wet summer to 3600 m$^3$/s in the dry winter) into the northern South China Sea through eight outlets (Figure 1). Four of the eight outlets (Humen, Jiaomen, Hongqimen, and Hengmen) enter Lingdingyang Bay, which has a trumpet-like shape with a width of 5 km near the northern end and 35 km at the southern end. Among the four outlets, the Humen Estuary is the largest mouth with a large amount of tidal influx and outflux between the estuary and offshore. It is obvious that the trumpet-like bay tends to amplify the tide amplitude in conjunction with the convergent estuary when the tidal wave propagates from the deep sea into the Humen Estuary in the landward direction. Thus, we focus on the characteristics of the tidal wave along the Humen Estuary, specifically the segment from the mouth upstream to approximately 1.5 km downstream from Huangpu Bridge.

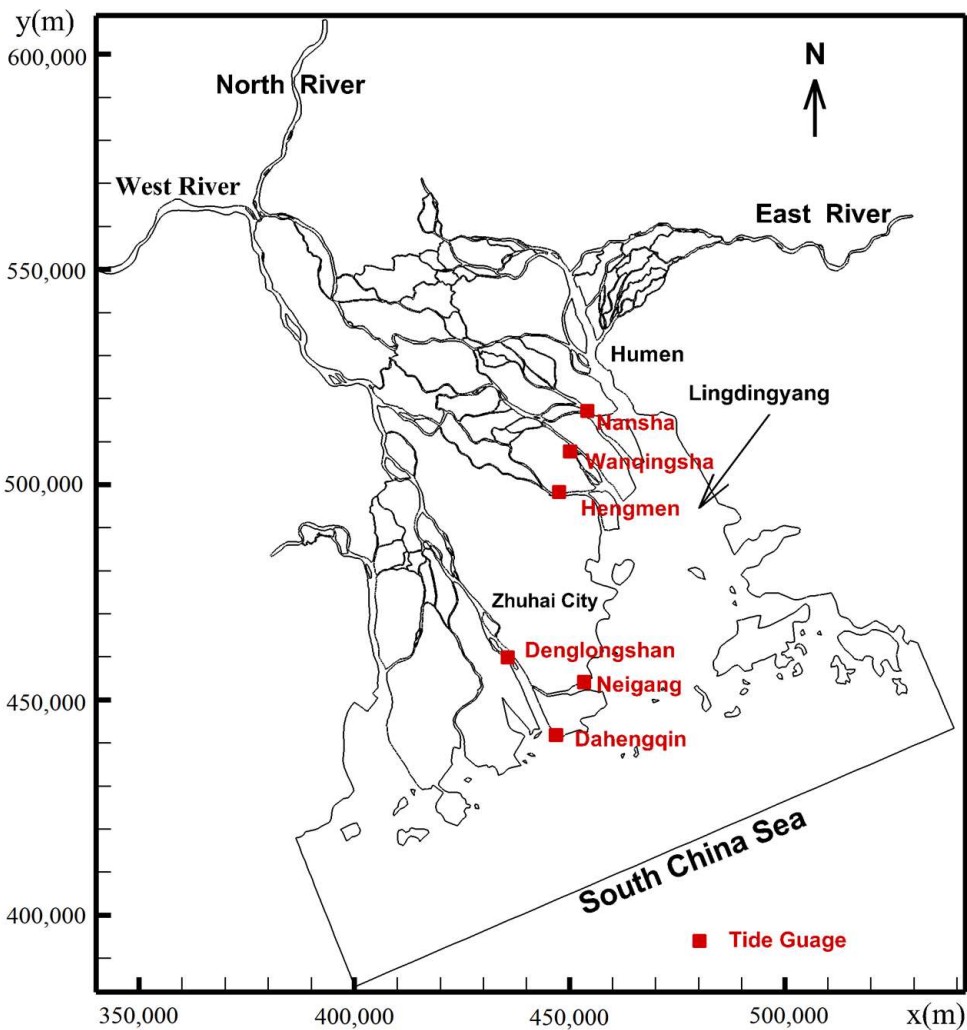

**Figure 1.** The location of the outlets of the Pearl River network.

### 2.2. Shape of the Humen Estuary

The shapes of most alluvial estuaries are similar all over the world. The width and the area of the cross section decreases in the upstream direction, resulting in a convergent

(funnel-shaped) estuary. The main geometric parameters of the convergent estuary can be described by exponential functions along the estuary axis with the origin at the mouth:

$$A = A_0 exp(-\frac{x}{a})$$
(1)

$$B = B_0 exp(-\frac{x}{b})$$
(2)

$$h = h_0 exp(-\frac{x}{d})$$
(3)

where $A$, $B$, and $h$ are the tidal average cross-sectional area, width, and depth; $A_0$, $B_0$, and $h_0$ are the same variables at the estuary mouth; $x$ is the distance from the estuary mouth; and $a$, $b$, and $d$ are convergent lengths of the cross-sectional area, width, and depth.

The parameters of Equations (1)–(3) can be obtained by regression on the topographic data. From the digital elevation model (DEM) of the Pearl River Delta, 54 cross sections were extracted to obtain the cross-sectional area, width, and depth (Figure 2). Then, the data were used to derive the convergent lengths $a$, $b$, and $d$ listed in Table 1. Figure 3 shows the regression lines plotted on semilogarithmic coordinates. It shows that the Humen Estuary can be divided into three sections: the downstream mouth section ($x$ = 0–8.7 km), the middle section ($x$ = 8.7–30.4 km), and the upstream section ($x$ = 30.4–36 km). The mouth section is near a prismatic estuary, and the middle and upstream sections are the typical convergent estuary.

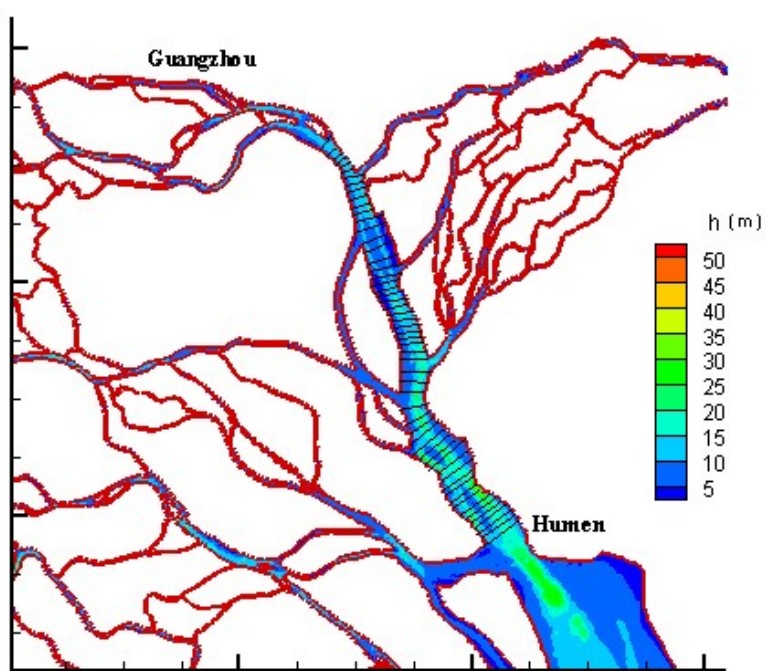

**Figure 2.** The deployment of the cross sections along the Humen Estuary.

**Table 1.** Shape characteristics of the Humen Estuary.

| Subsections | Range (km) | A (km) | B (km) | D (km) |
|---|---|---|---|---|
| Mouth | 0–8.7 | 166.7 | 333.3 | 333.3 |
| Middle | 8.7–30.4 | 25 | 71.4 | 40.0 |
| Upstream | 30.4–36 | 20 | 8.3 | 208.3 |

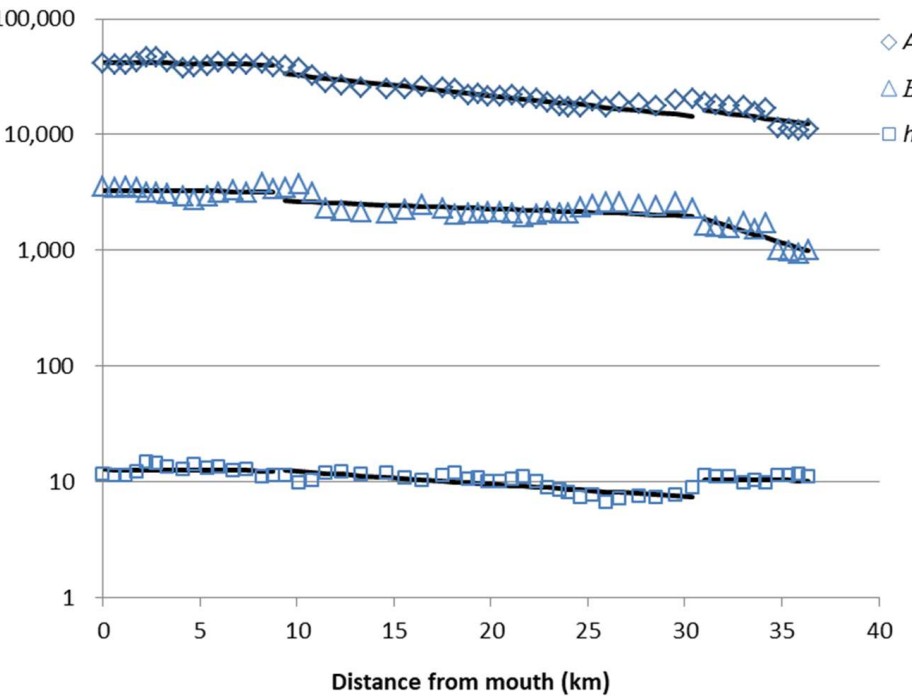

**Figure 3.** The regression lines for the cross-sectional area $A$ (m$^2$), width $B$ (m), and depth $h$ (m) along the estuary.

## 3. Analytical Solution

The tidal dynamics in an estuary can be described by the following Saint–Venant equations:

$$\frac{\partial U}{\partial t} + U\frac{\partial U}{\partial x} + g\frac{\partial d}{\partial x} + gI_b + gn^2\frac{U|U|}{d^{4/3}} = 0 \tag{4}$$

$$r_s\frac{\partial A}{\partial t} + \frac{\partial Q}{\partial x} = 0 \tag{5}$$

where $U$ is the tidal flow cross-sectional velocity, $g$ is the acceleration due to gravity, $d$ is the flow depth, $I_b$ is the bottom slope, $n$ is the Manning coefficient, and $Q$ is the tidal flow discharge. $r_s$ is defined as the ratio between the storage width and the stream width, which is often more than unity in the river with shallow plains [14] (Savenije, 2008).

After the scaling of Equations (4) and (5) and under the assumption of a harmonic wave traveling from the estuary mouth to the upstream reach, the following dimensionless parameter can be obtained:

$$\gamma = \frac{c_0}{\omega a} \tag{6}$$

$$\chi = r_s f\frac{c_0}{\omega h}\zeta \tag{7}$$

$$\mu = \frac{1}{r_s}\frac{vh}{\eta c_0} \tag{8}$$

$$\lambda = \frac{c_0}{c} \tag{9}$$

$$\delta = \frac{1}{\eta}\frac{d\eta}{dx}\frac{c_0}{\omega} \tag{10}$$

where $c_0$ is the classical wave celerity of a frictionless progressive wave, $\omega$ is the frequency of the wave, $\zeta = \frac{\eta}{h}$ is the dimensionless tidal amplitude, and $f$ is the dimensionless friction factor defined as

$$f = \frac{gn^2}{h^{1/3}} \left[ 1 - \left( \frac{4}{3} \zeta \right)^2 \right]^{-1} \tag{11}$$

Using the above dimensionless parameters in Equations (6)–(11), the scaling form of Equations (4) and (5) can be solved on the basis of the equations for the phase lag:

$$\tan(\epsilon) = \frac{\lambda}{\gamma - \delta} \tag{12}$$

$$\mu = \frac{sin(\epsilon)}{\lambda} = \frac{cos(\epsilon)}{\gamma - \delta} \tag{13}$$

$$\delta = \frac{\mu^2}{\mu^2 + 1}(\gamma - \chi\mu^2\lambda^2) \tag{14}$$

$$\lambda^2 = 1 - \delta(\gamma - \delta) \tag{15}$$

where $\epsilon$ is the phase difference between high water and high water slack. It is clear that, for a progressive wave, $\epsilon = 0.5\pi$ and that, for a standing wave, $\epsilon = 0$. The above equations should be applied to the following conditions: (1) the ratio between the tidal amplitude and the water depth is less than unity, (2) the runoff is much less than the tidal influx and outflux, and (3) the incident wave can be simplified as a harmonic wave. Equations (12)–(15) can be solved by an iteration process when $\gamma$ and $\chi$ have been determined.

## 4. Numerical Model and Verification

The Finite Volume Community Ocean Model (FVCOM), an unstructured-grid finite-volume, three-dimensional primitive equation coastal ocean model developed originally by Chen et al. [19] and upgraded by the UMASS-D/WHOI model development team [20,21], is applied to simulate the tide and surge propagation process in the Pearl River network. As shown in Figure 4, the model domain covers the entire region of the Pearl River network, eight estuaries, and the nearshore. The model contains 58,564 nodes and 91,572 elements. Horizontally, the spatial resolution varies from 40 to 50 m within the river network and from 600 to 1000 m near the estuarine mouth and bay to 5 km at the offshore open boundary. To model the surge under various approaching cyclones in summer days, the Pearl River network model is nested at the open boundary with the South China Sea model, which is also based on the spherical coordinate version of the FVCOM. Vertically, the model contains five uniform sigma layers to capture the process of turbulence momentum transferring upwards from the bottom.

The forces driving the model include tidal waves from the South China Sea and surges caused by meteorological forces such as storms, cyclones, and rainfalls. In this paper, we consider only tropical cyclone. The semi-empirical parametric cyclone model is integrated into the ocean model to drive the surge. The advantage of the parametric cyclone model is that it does not require as much meteorological data as the dynamic model to drive the model. The only data needed contain the track of the cyclone, the center pressure drop, the forward speed, and the maximum wind radius, most of which are available on weather forecast websites except for the maximum wind radius, which should be obtained using a statistical formula and then adjusted according to the surge results. Here, the cyclone No.1822 (Table 2), well known as the Super Typhoon Mangosteen (2018), is used as an example to simulate the storm surge in conjunction with the astronomical tide. The numerical results are compared with the measurements and demonstrate good agreement in the high surge level between them in Figure 5.

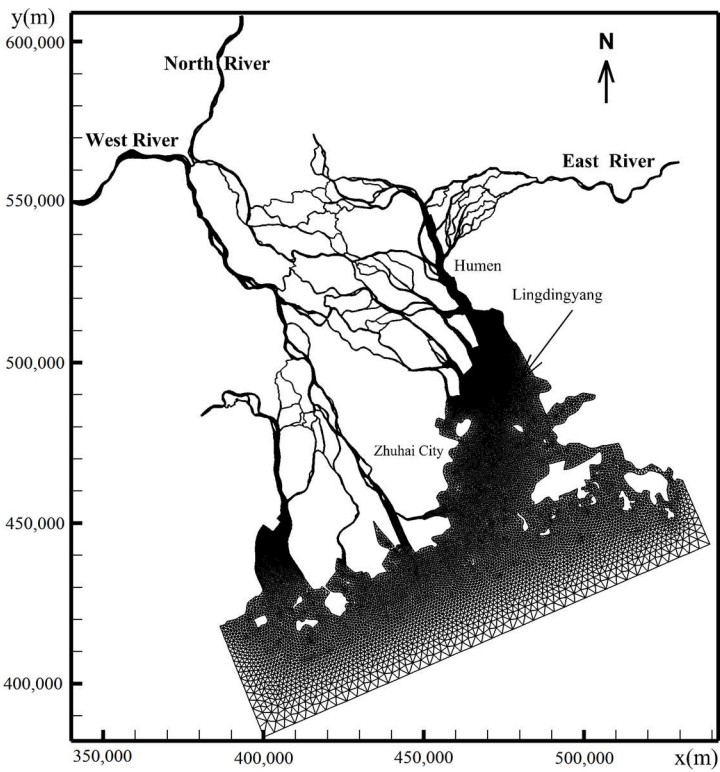

**Figure 4.** The domain and grids of the Pearl River network model.

**Table 2.** The parameters of the Typhoon Mangosteen (two days before landfall).

| Time | Latitude (N) | Longitude (E) | Pressure (hPa) | Maximum Wind Speed (m/s) |
|---|---|---|---|---|
| 00:00 14/09 | 16 | 126.9 | 905 | 56.6 |
| 06:00 14/09 | 16.7 | 125.7 | 905 | 56.6 |
| 12:00 14/09 | 17.4 | 124.1 | 905 | 56.6 |
| 18:00 14/09 | 18 | 122.3 | 905 | 56.6 |
| 00:00 15/09 | 18 | 120.5 | 940 | 46.3 |
| 06:00 15/09 | 18.5 | 119.7 | 940 | 46.3 |
| 12:00 15/09 | 19.2 | 118.3 | 950 | 43.7 |
| 18:00 15/09 | 19.8 | 117 | 955 | 41.2 |
| 00:00 16/09 | 20.6 | 115.3 | 960 | 38.6 |
| 06:00 16/09 | 21.7 | 113.5 | 960 | 38.6 |
| 12:00 16/09 | 22.2 | 111.6 | 970 | 33.4 |
| 18:00 16/09 | 22.7 | 109.7 | 980 | 28.3 |

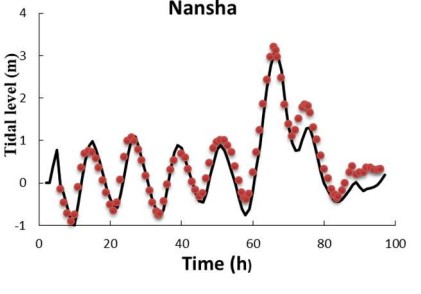

(**a**) Nansha Station　　　　　　　　　　　(**b**) Wanqingsha Station

**Figure 5.** *Cont.*

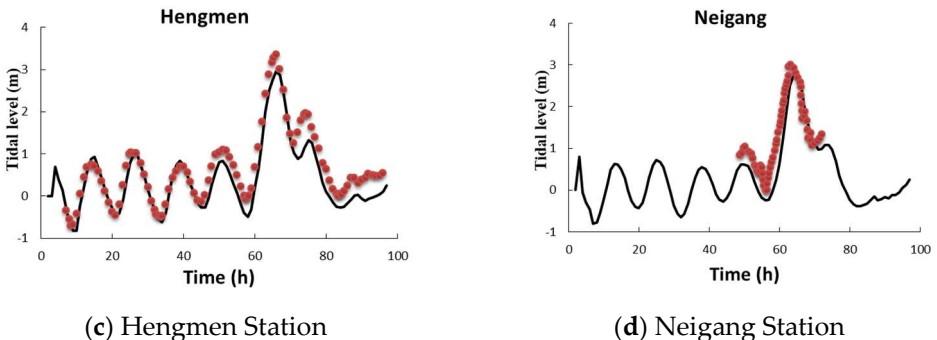

(**c**) Hengmen Station            (**d**) Neigang Station

**Figure 5.** Verification of the numerical model in four stations: Nansha, Wanqinsha, Henmeng, and Neigang.

## 5. Results

We used the Humen Estuary geometry presented in Table 1 and the numerical results on 8–9 September 2018 (about one week before the typhoon landfall) to calibrate the analytical model. The calibration process are as follows: we first divided the parameter range into small subareas with an increment of 0.1 for $r_s$ and 0.001 for $n$ with their ranges listed in Table 3. The range of $r_s$ and $n$ were referred to Savenije et al. (2005; 2008) [1,14]. Then, we used these parameters to obtain the tidal results. Lastly, we compare the analytical solutions with the numerical results and compute the RMSE (root mean square error) for every analytical solution corresponding to the parameters. The one with the least RMSE is considered the best fit solution, and its parameters are the optimal ones. The calibration parameters are presented in Table 3. In general, the analytical solution fits the numerical result very well (Figure 6), and the calibrated parameters are within the physically proper range. In contrast, in the upstream section, there appears to be more deviation than in the other two sections. The reason is probably that the length of this section is too short to match a more reasonable convergent length.

**Table 3.** Parameters used for analytical model.

| Subsections | Storage Width Ratio $r_s$ | Value Range | Manning's Coefficient $n$ ($\mathrm{m^{-1/3}s}$) | | |
| --- | --- | --- | --- | --- | --- |
| | | | Astronomical Spring Tide | Considering Cyclone Surge | Value Range |
| Mouth | 1 | 1–2 | 0.005 | 0.005 | 0.017–0.06 |
| Middle | 1.8 | 1–2 | 0.031 | 0.018 | 0.017–0.06 |
| Upstream | 1.5 | 1–2 | 0.035 | 0.015 | 0.017–0.06 |

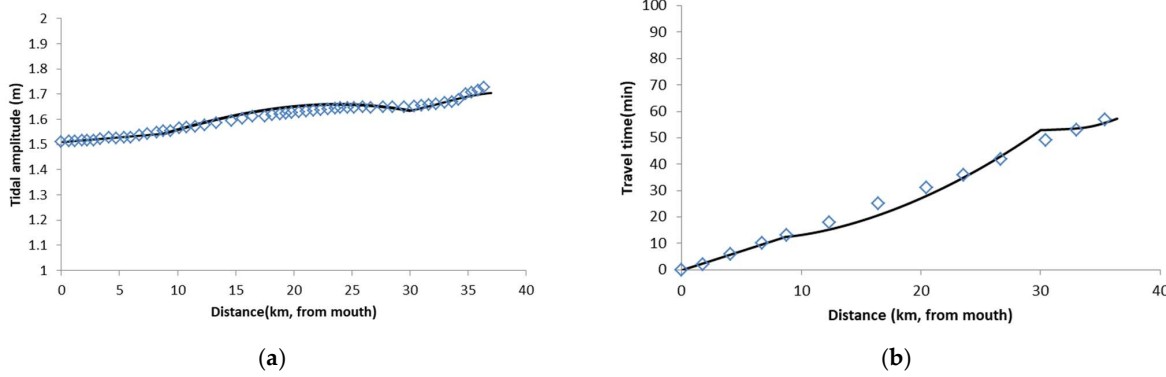

(**a**)                            (**b**)

**Figure 6.** Comparison of the analytical results for (**a**) tidal amplitude and (**b**) travel time, and the numerical simulation after calibration during 8–9 September 2018. The diamonds denotes the numerical simulation results and the same for the following figures.

The model was further verified with the numerical results on 16–17 September 2018. The reason for choosing this period is that Super Typhoon Mangosteen approached and made landfall on the coast of Guangdong Province during this time. Thus, at first, we only simulated the astronomical tide without exerting a wind force in the numerical model. Through this simulation, the analytical result was compared with the numerical result assuming no cyclone landfall. From Figure 7, it can be seen that the correspondence with the numerical result is good for the tidal amplitude. However, the deviation for the travel time shows that the analytical model tends to overestimate the celerity compared with the numerical model. A similar problem concerning the distinct accuracies between the tidal amplitude and the travel time was also encountered by others using Savenije's analytical model, who attributed this deviation to the effect of the storage width ratio $r_s$ at different tidal amplitudes (Cai et al., 2012).

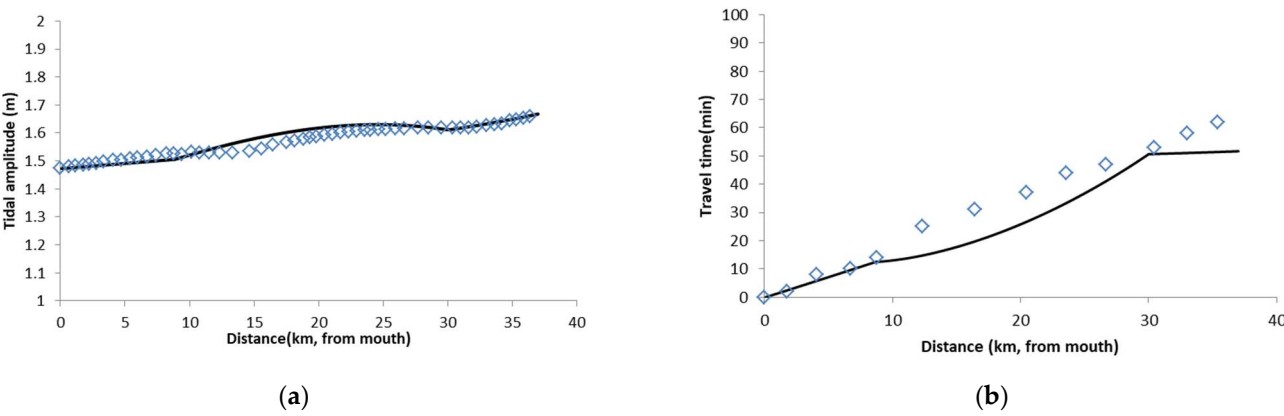

(**a**)    (**b**)

**Figure 7.** Comparison of the analytical results for (**a**) tidal amplitude and (**b**) travel time, and the numerical simulation for verification under astronomical tide during 16–17 September 2018.

Then, we simulated the typhoon-surge scenario by setting the cyclone model and by exerting the wind and pressure forces on the surface of the tidal flow. The analytical result was compared with the numerical result. It is shown in Figure 8 that the difference between them can reach a maximum of as much as 0.6 m. This result is unsurprising because the original scope of the analytical model does not include storm surges. With respect to the equation, a wind stress term should be supplemented in the momentum equation as follows:

$$\frac{\partial U}{\partial t} + U\frac{\partial U}{\partial x} + g\frac{\partial d}{\partial x} + gI_b + gn^2\frac{U|U|}{d^{\frac{4}{3}}} - \tau_W = 0 \qquad (16)$$

where $\tau_W$ is the wind stress along the x-coordinate direction with a positive direction from the mouth to the upstream.

From Equation (16), we can find that the effect of the positive wind stress term, which caused a maximum surge, is equivalent to a reduction in the Manning coefficient:

$$gn^2\frac{U|U|}{d^{\frac{4}{3}}} - \tau_W = gn_W^2\frac{U|U|}{d^{\frac{4}{3}}} \qquad (17)$$

$$n_w = \sqrt{n^2 - \tau_W\frac{d^{\frac{4}{3}}}{gU|U|}} \qquad (18)$$

where $n_w$ is considered the reduced version of the original Manning coefficient $n$ under the cyclone surge scenarios.

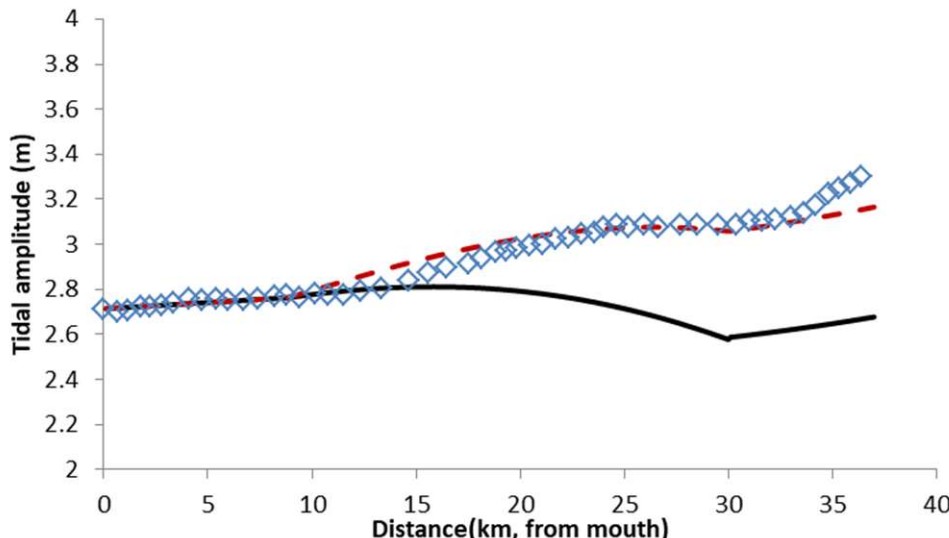

**Figure 8.** Comparison of the analytical tidal amplitude before (solid line) and after (dash line) tuning Manning's coefficient, and numerical simulation for verification under tide and surge during 16–17 September 2018.

Therefore, we can make the analytical solution close to the numerical result by tuning Manning's coefficient to be less. The analytically calculated tidal amplitudes before and after tuning the coefficient are shown in Figure 8, and all of the parameters are listed in Table 3. The Manning coefficient considering the cyclone surge is significantly less than that only considering the astronomical tide. Some are even below the lower limit of the physically reasonable range. This means that the effect of the storm surge on the tidal damping is similar to the reduction in the bottom friction. In other words, the wind that blew in the upstream direction before landfall of the cyclone counteracted part of the friction when the tidal wave traveled upstream along the river. For an application of the solution in a surge scenario and a physically reasonable estimate of the Manning coefficient in the meantime, a new solution based on Savenije's analytical model, which contains the wind stress term, should be derived in the future.

## 6. Discussion

### 6.1. The Characteristics of the Tidal Wave Propagation in the Humen Estuary

Savenije's solution describes the variation of the four dimensionless parameters $\epsilon$, $\mu$, $\delta$, and $\lambda$ as a function of the shape number $\gamma$ and the friction number $\chi$, which is depicted in Figure 9. As shown in these figures, the solutions can be generally divided into two families. For the areas with a small shape number $\gamma$ ($\gamma < 2$), all of the parameters depend on friction number $\chi$. The phase lag $\epsilon$ decreases with an increase in $\chi$, and the tidal wave is a mixture of the progressive wave and standing wave, indicating a riverine estuary. In contrast, for the areas with a large shape number $\gamma$ ($\gamma > 3.5$), the friction number has little influence on the four dimensionless parameters. The phase lag $\epsilon$ increases with an increase of $\chi$, and the tidal wave is much closer to the standing wave, indicating an oceanic estuary. In the intermediate area within $2 < \gamma < 3.5$, the critical $\gamma$ for transforming from the progressive wave to the standing wave increases with the increase in $\chi$.

The hydrodynamic analysis, as with the geometric analysis, indicates that there are three distinct segments for the mainstream of the Humen estuary. Figure 9 shows the parameters for the three segments of the estuary. In the mouth section (0–8.7 km), as a result of the nearly prismatic estuary, this section is featured by the riverine estuary due to the small $\gamma$. Moreover, the friction in this section is also very small. Both the small $\gamma$ and $\chi$ make the tidal wave in this section travel as a series of progressive waves, with small amplification of the wave amplitude in the upstream direction. In the middle section

(8.7–30.4 km), $\gamma$ is within the intermediate area and becomes small upstream while $\chi$ becomes large in the same direction. The sectional curve is crossed and divided into two parts by the ideal river curve. Above the ideal river curve, the effect of cross-sectional convergence dominates over the friction; thus, the tidal energy is accumulated and the tide is amplified in the downstream part of the middle section. Below the ideal river curve, the friction effect dominates over the cross-sectional convergence; thus, the tidal energy is dissipated and the tide is damped. The analysis corresponds with the result in the last section. In the upstream section (30.4–36 km), the large $\gamma$ makes this section more similar to an oceanic estuary. The convergence effect is superior to friction according to the value of $\delta$; thus, the tide is amplified. The wave in this section is typical of a standing wave, which is nearly independent to the friction. In the process of calibration, this section is insensitive to Manning's coefficient.

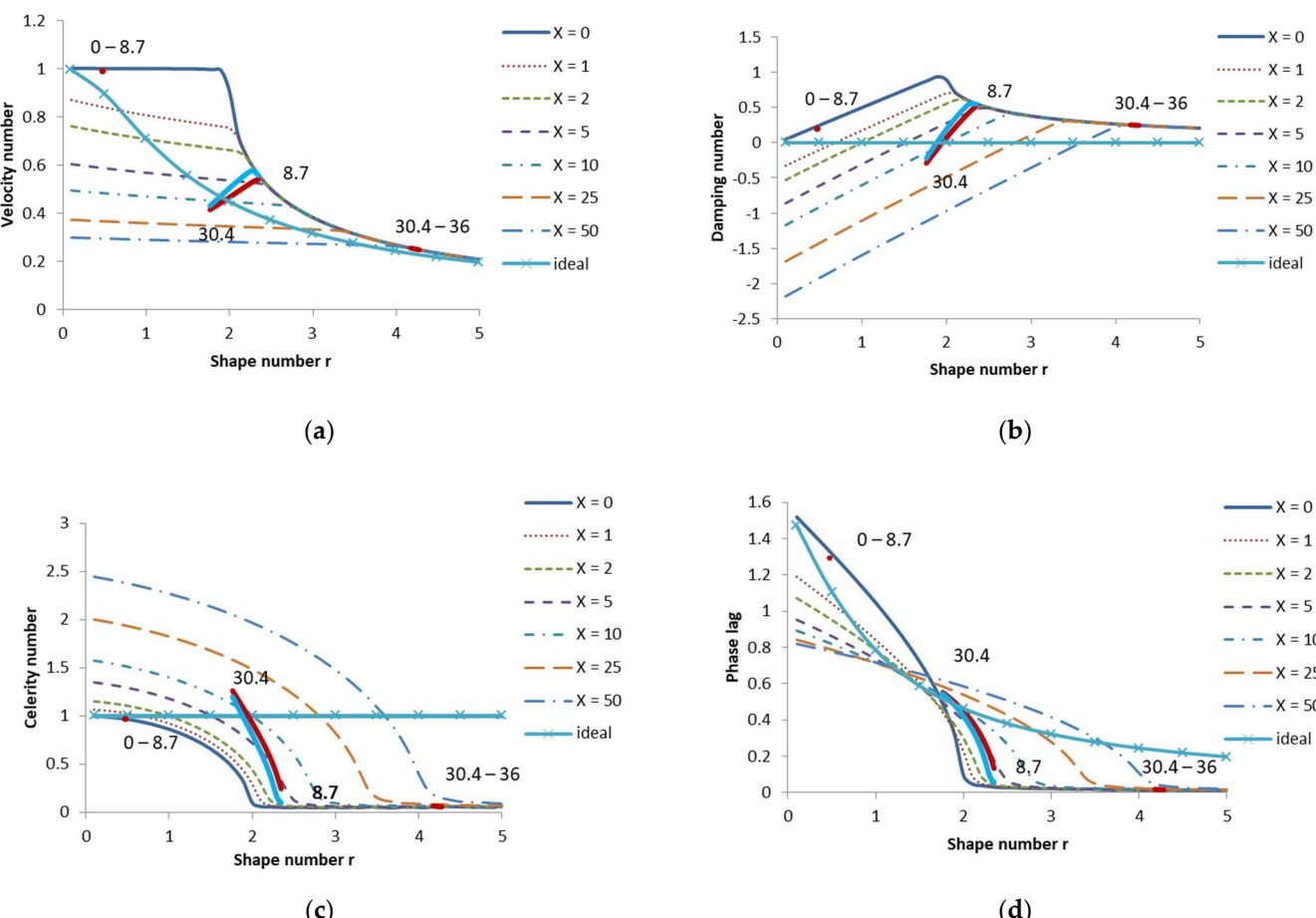

**Figure 9.** The relation between four parameters: (**a**) velocity number, (**b**) damping number, (**c**) celerity number, and (**d**) phase lag. The estuary shape number $\gamma$ for different friction numbers $\chi$ is indicated by different line types. The bold lines indicate the parameters for the Humen Estuary under astronomical tide (red) and tide plus surge (blue), with the numbers indicating the distance from the estuary mouth in kilometers. The drawn line with the crosses represents the ideal estuary.

To investigate the difference made by the storm surge compared with the normal tidal wave, we depicted the calibrated parameter curves in the middle section under the storm surge. As expected, the curves under the surge move slightly toward the zone with less friction from the original curves under the astronomical tide. This is in agreement with the analysis of the results in the former section. Previous studies have indicated that runoff can increase the friction for wave propagation in an estuary; however, this study shows that storm surge can reduce the friction for the wave propagation while exaggerating the wave amplitude.

*6.2. Comparison with Other Estuaries*

The Humen Estuary is so complicated that it can be divided into three sections. Any section demonstrates its distinct features compared with the others. To better understand the property, we compared them with other typical estuaries, which have been analyzed by Savenije et al. [14] (2008). As shown in Figure 10, the feature of the mouth section (0–8.7 km) in the Humen estuary can be comparable with the Hau Estuariy, especially the upstream section (57–160 km). They are all riverine estuaries with small shape numbers. Thus, they have a riverine character with a long convergence length and a high phase lag. The difference is that the mouth section of the Humen Estuary has less friction than the Hau Estuary. Therefore, the tide is amplified for the former and damped for the latter, and the celerity in the mouth section of the Humen Estuary is larger than that of the Hau estuary. The middle section (8.7–30.4 km) of the Humen Estuary is comparable with the Schelde Estuary, with close shape numbers located in the intermediate zone. Both of their curves cross the ideal estuary curve, which indicates a transition point between tidal damping and amplification. The difference is that the depth of the Humen Estuary is decreased in the upstream direction and that it is the opposite for the Schelde Estuary. Moreover, the length of the middle Humen estuary is much less than that of the Schelde Estuary, but the curve length is similar. This means that the parameter variation rate of the middle Humen Estuary is greater than that of the Schelde Estuary.

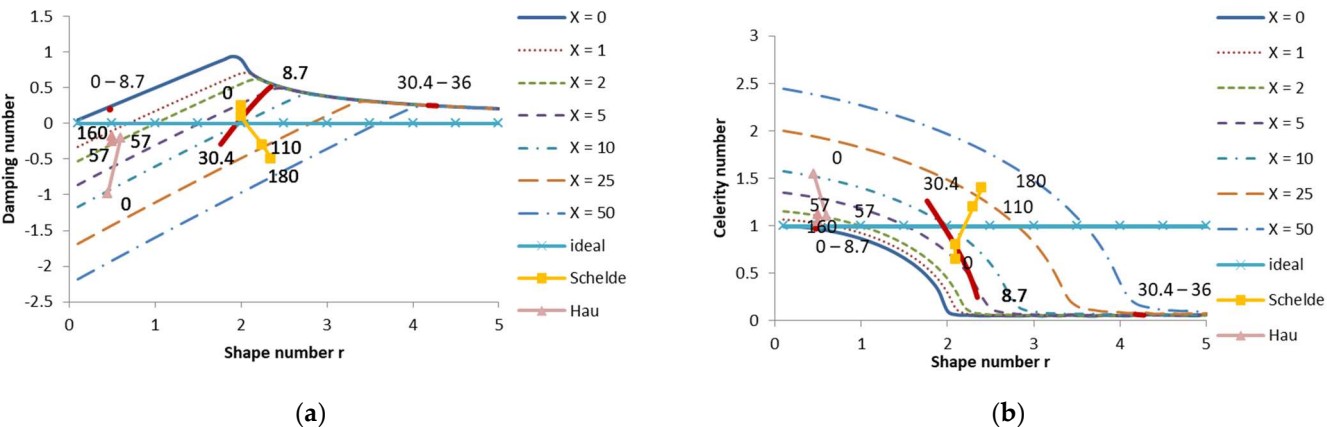

(**a**)  (**b**)

**Figure 10.** Positioning of the Humen (red line), Schelde (yellow squares), and Hau (pink triangles) Estuaries in the damping number (**a**) and celerity number (**b**) diagrams, with the numbers at the inflection points indicating the distance from the estuary mouth (in kilometers). The other lines are the same as those in Figure 9.

## 7. Conclusions

In this study, Savenije's solution was applied for the first time to the Humen Estuary. Due to the scarcity and low resolution of the measured data, we used the numerical results to calibrate and verify the analytical model. The merit of using an analytical model is that it can provide direct insight into relationships among the tidal properties, such as velocity amplitude, the tidal damping rate, and wave celerity; the geometry indicator; friction; and the tidal forces. This analytical model demonstrated good accuracy when it was applied to an estuary where the tide dominates in comparison with the river discharge. The results indicate that the analytical model can predict the astronomical tide well in the Humen Estuary after calibration. However, it cannot predict the tide well when a tropical cyclone-induced surge is superimposed onto the astronomical tide. The reason may lie in the fact that Savenije's solution does not take the wind forces into account. After reducing Manning's coefficient, we found that the analytical result could be close to the numerical results. This means that the loss of the wind forcing can partly be compensated by adjusting the friction.

By analyzing the tidal wave propagation along the Humen Estuary, we found that the characteristics of the three sections—the mouth section (0–8.7 km), the middle section (8.7–30.4 km), and the upstream section (30.4–36 km)—are not alike at all. The mouth section is a typical riverine estuary with a nearly constant cross section. The tidal wave traveling there is in the form of a progressive wave and with a nearly classical celerity in a frictionless state. In contrast, the upstream section is a typical oceanic estuary with a short convergence length. The tidal wave there is in the form of a standing wave with a nearly infinite celerity. The middle section is in the intermediate zone of the two solution families. The tidal wave is first amplified and then damped, with a transition point that is the location of the maximum tidal amplitude. Finally, a comparison was conducted between the Humen Estuary and other estuaries.

**Author Contributions:** Conceptualization, Z.Z. and F.G.; methodology, Z.Z.; validation, Z.Z.; formal analysis, D.Z.; investigation, Z.Z.; resources, D.H.; data curation, D.H.; writing—original draft preparation, Z.Z.; writing—review and editing, F.G.; visualization, D.Z.; supervision, D.Z.; project administration, F.G.; funding acquisition, F.G. All authors have read and agreed to the published version of the manuscript.

**Funding:** This paper was supported by the National Key R&D Program of China (grant No. 2018YFB0505500 and 2018YFB0505502) and by the National Natural Science Foundation of China (grant No. 41771421, 41771447, and 41571386).

**Institutional Review Board Statement:** Not applicable.

**Informed Consent Statement:** Not applicable.

**Data Availability Statement:** No report any data.

**Conflicts of Interest:** The authors declare no conflict of interest.

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
