# Peer review of "An Analytical Solution for Investigating the Characteristics of Tidal Wave and Surge Propagation Associated with Non-Tropical and Tropical Cyclones in the Humen Estuary, Pearl River"

_water, doi:10.3390/w13172375_

Round 1
Reviewer 1 Report
The article describes the characteristics of the Humen estuary and some of the modeling considerations that are associated with the region. Given the importance of the region, this is a valuable contribution to the literature. The article is focused on only the tidal contributions and how to develop an analytical model of the estuary. This is perhaps the most confusing thing about the article, surge propagation is done numerically but then ignored beyond calibration. To say the least, this is confusing to the reader and either should be removed or clearly explained. There are several issues with modeling the surge beyond this that are not addressed, but removing this would address this problem.
Beyond the issues above, the analytical model being used is okay, but there is no mention of how the analytical parameters are matched to the domain. This is a traditional inverse parameter fitting problem for which there are several ways to approach. Explaining the approach to the fitting would be of great interest not only for others attempting to do something similar, but also for reproducibility’s sake.
It is also puzzling as to why the authors created an FVCOM model of the estuary and then suggest that it is not adequate, at least implicitly. One would assume that the FVCOM model would be excellent at reproducing the complexity of the estuary, but this seems to be discarded. There are benefits of having an analytical representation of the estuary, but discounting the numerical result seems unwise. Instead, why one would not want to validate the numerical solution seems a missed opportunity.
# Comments (page, line number)
- (2, 11) Numerical models can provide direct insight into the physics of any estuary, arguably better than any other method. This statement beyond being misleading also seems to motivate the use of an analytical model. While an analytical model still has merits, justifying it this way is incorrect.
- (6, 9-25) Wind intensity is not mentioned and one would assume part of the reconstruction
- (6, 9-25) What parameterization are you using to reconstruct the storm?
- (6, 9-25) The domain is too small to really capture a storm surge. Has there been any effort to test larger domains to see the difference in numerical results?
- The results in figure 5 actually look very good! I am not sure if I understand why given these results that the comments in the paragraph above are so disparaging.
- (7, 15-16) What does the statement "The numerical model can change its meteorological conditions according to research" mean?
- (8, 1) "precisions" here is probably meant to be accuracy?
- Many of the figures could be replaced instead by misfitting of the model or analytical solution.
- (11, 1) What does it mean for the "The convergence effect is superior to the friction"? It is not surprising that the friction is a dominant parameter, but saying it is independent seems to suggest something else.
- (13, 25) The statement here that the analytic solution does not consider the wind is superfluous. Why state that something that does not consider a surge and then say again it's no good because there's wind forcing?
- (14, 2) "This means the loss of the wind forcing can partly be compensated by adjustment of the friction forcing." Is this statement supposed to imply that storm surge can be represented by friction adjustments? This is clearly untrue, but what was the intent of this statement?
Reviewer 2 Report
Overall as I am satisfied with content of this article and research approach. I would like to recommend this article to be successfully published in our journal.
I would like to recommend authors to revise this article considering a few following suggestions as below:
(1) Requirement
1) In page 1, as your analytical model was applied for tidal wave and surge by tuning the Manning's coefficient for cyclone surge, a slight modification of the tile may be better.
An analytical solution for investigating the characteristics of tidal wave and surge propagation in Humen Estuary, Pearl River
--> An analytical solution for investigating the characteristics of tidal wave and surge propagation associated with non-and tropical cyclones in Humen Estuary, Pearl River
2) Abstract is too long.
on Line 11~13--- Rewrite clearly by a direct expression in English,
(However, little research has quantified the hydraulic process between the geometry, topography, and tidal wave and surge, not to mention the subdivision of the reaches based on mechanics with respect to this estuary.)
3) On 23 ~ 25 in abstract, please omit this sentence.
4) In sentences from 18 ~25 in page 6, I think that numerical simulation results in Fig. 5 is quite good. I don't agree that "Some discrepancies for low surge may be caused by -------cyclone model", because we can not simply say, due to different result from different conditions indicated above.
Please omit " Considering that this ---- in an estuary, this error is acceptable".
5) Number of Typhoon Mangosteen (2018) is 1822?
According to "Typhoon season in 2018", NO. 1822 is Tropical storm "Rumbia" whose track passed by Shanghai.
or No 1827 is Typhoon "Mangkhut (Ompong)" whose track passed by near Gangzhou.
Please check it again.
6) As shown in Fig. 8, your approach is quite good using Eqs 16, 17 and 18 for considering "positive wind stress caused maximum surge being equivalent to a reduction of the Manning coefficient. However, simply the adjustment of the Manning coefficient for wind stress will be limited in successfully producing in the different cases of typhoon.
As actually the calculation of wind stress is not difficult and typhoon effect on tide and tidal wave is really big, I recommend that you can just put more detailed wind stress term in your analytical model, referring to most of numerical models for tropical cyclone as an easy way.
(2) Through minor correction, you can complete a final paper before its publication.
I expect your future research to be more successful.
Author Response
Dear Reviewer:
We sincerely thank the editor and reviewers for their valuable suggestions that would assist in the improvement of the quality for our manuscript. The reviewer comments are laid out below one by one and the point-to-point responses are just followed the comments. Note that the page and line numbers are corresponding to the original paper but not the revised one. Specifically,
1 In page 1, as your analytical model was applied for tidal wave and surge by tuning the Manning's coefficient for cyclone surge, a slight modification of the title may be better.
An analytical solution for investigating the characteristics of tidal wave and surge propagation in Humen Estuary, Pearl River
--> An analytical solution for investigating the characteristics of tidal wave and surge propagation associated with non-and tropical cyclones in Humen Estuary, Pearl River
Response 1:
I agree to modify this title into ‘An analytical solution for investigating the characteristics of tidal wave and surge propagation associated with non-and tropical cyclones in Humen Estuary, Pearl River’. That is more specific than the original one. Thank the reviewer for patience.
2 Abstract is too long.
on Line 11~13--- Rewrite clearly by a direct expression in English,
(However, little research has quantified the hydraulic process between the geometry, topography, and tidal wave and surge, not to mention the subdivision of the reaches based on mechanics with respect to this estuary.)
3 On 23 ~ 25 in abstract, please omit this sentence.
Response 2 and 3:
I have changed the sentence (Line 11-13) into a direct expression as:
‘However, little research has quantified the relationship between the hydraulic characteristics and the geometry features in this estuary.’
However, I don’t think it is appropriate to delete the last two sentences (Line 23-25) in Abstract. Because I have analyzed the characteristics of the Humen Estuary in Discussion from Page 10-13 and in Fig. 9 and Fig. 10. This should be mentioned in Abstract. So, I change the original two sentences into the following concise one as:
‘Finally, we analyzed the characteristics of the tidal wave in Humen Estuary using the analytical solution and its parameters’.
4 In sentences from 18 ~25 in page 6, I think that numerical simulation results in Fig. 5 is quite good. I don't agree that "Some discrepancies for low surge may be caused by -------cyclone model", because we can not simply say, due to different result from different conditions indicated above.
Please omit " Considering that this ---- in an estuary, this error is acceptable".
Response 4:
Okay, since two reviewers both think the numerical simulation results are good, we OMIT the above sentences for a concise expression.
5 Number of Typhoon Mangosteen (2018) is 1822?
According to "Typhoon season in 2018", NO. 1822 is Tropical storm "Rumbia" whose track passed by Shanghai.
or No 1827 is Typhoon "Mangkhut (Ompong)" whose track passed by near Gangzhou.
Please check it again.
Response 5
We have checked the typhoon Mangosteen (2018), the No. 1822 is right. The Tropical storm Rumbia (2018) is identified by No. 1818. We used the Tropical Cyclone Number according to Chinese Central Meteorological Station. Maybe in other countries are different.
6 As shown in Fig. 8, your approach is quite good using Eqs 16, 17 and 18 for considering "positive wind stress caused maximum surge being equivalent to a reduction of the Manning coefficient. However, simply the adjustment of the Manning coefficient for wind stress will be limited in successfully producing in the different cases of typhoon.
As actually the calculation of wind stress is not difficult and typhoon effect on tide and tidal wave is really big, I recommend that you can just put more detailed wind stress term in your analytical model, referring to most of numerical models for tropical cyclone as an easy way.
Response 6
Yes, we agree with the suggestion that the wind forcing should be included in the solution when it is applied in a context of surge with tide. In this paper, we only use Eq.(18) to adjust the physical Manning coefficient to enlarge the scope of application for Savenije’s analytical solution at the current version. We already have this idea but the derivation of the revised Savenije’s solution is very trivial and some problems should be fixed firstly. We think this can be completed in out next publication. Thanks for your good suggestion!
Round 2
Reviewer 1 Report
# An analytical solution for investigating the characteristics of tidal wave and surge propagation associated with non-and tropical cyclones in Humen Estuary, Pearl River
## Summary
The authors have addressed many of the concerns with the article. Some
additional comments and responses are listed below. The primary issue still
existing is the use of an analytical model that does not take into account surge
input. While I can understand the interest in creating/evaluating an
analytically simple model it is still confusing. I suggest making it very
clear, perhaps even enumerating clearly the goals of the article explicitly, so
that there is not a misinterpretation into what the article is reporting.
## Notes
- Understood that the article does not only focus on the tidal component and
that the surge component is being considered, it is just confusing as to how
the analytical solution and numerical solution that includes surge should be
compared. Response 1 does give a bit more clarity but I believe that the
goal as quoted should be made much more clearly, even in the abstract as it
is an important goal of the article.
- Validating the analytical model using the numerical model is a fine procedure
although there can be arguments against it given that it does assume that the
numerical representation is validated itself. I believe that you did this to
some extent, although more validation would have been nice.
- Regarding the inverse problem approach what I am referring to is describing
how you searched the parameter space. This space can be high-dimensional.
If you simply evaluated the parameter space in a uniform, tensorial way then
I would state that. The resolution of this evaluation though is important
and I did not see that being mentioned. Also just saying that you are doing
a least-squares fit would suffice if you would like.
- The idea that analytical solutions, or I would say a "surrogate" or "reduced
order model" can be extremely helpful is definitely something that is well
accepted. I might suggest using some of this terminology to get a wider
understanding of what it is you are doing given that this is effectively what
you are doing.
- Regarding response 4, I believe we agree but I would disagree still with
the statement that "...numerical models do not provide direct insight...".
Instead what I think you are saying is that one can extract data from a
numerical model that can provide direct insight. In some sense your
analytical model is building a regression model of the numerical solution and
therefore is a direct result of the numerical model, which contains the
insight, it just needs extraction from the resulting data.
- Regarding response 6, I still believe you are being harsh but feel free to be
careful. I know that many in the storm surge modeling community would be
happy with this result.
- Regarding response 7, I am not sure that it makes sense to state that "the
meteorological conditions" can change. I believe this is an assumed
capability for any numerical model and only lead to confusion.
- Regarding response 9, I would just make it clear in the article that this is
what you are referring to.
- Regarding response 10 + 11, you have identified my primary concern with the
goals as stated in the article. The goals of the paper seem to me a bit
contradictory due to this. I agree that it is interesting to think about the
analytical model as perhaps a means for deriving further insight but without
the processes involved it's difficult to see how this makes sense.
Author Response
## Summary
The authors have addressed many of the concerns with the article. Some
additional comments and responses are listed below. The primary issue still
existing is the use of an analytical model that does not take into account surge
input. While I can understand the interest in creating/evaluating an
analytically simple model it is still confusing. I suggest making it very
clear, perhaps even enumerating clearly the goals of the article explicitly, so
that there is not a misinterpretation into what the article is reporting.
Response:
To be exact, the analytical model does not take into account wind stress so we guess (not sure) it may not be fit for surge. Because the surge has been formed driven by the remote wind at the estuary mouth and then propagate upstream. From the physical viewpoint, the surge has been formed as a series of gravity waves as well as the astronomical tide waves. The analytical solution can capture these free waves well if the local wind forcing was insignificant (but not sure), because the typhoon has made landfall and was weakening at that time, and the direction of the wind may not be parallel to the stream. Based on the above considerations, we want to test the analytical model using in the context of surge and tide.
We have rewritten the goals of the article.
## Notes
- Understood that the article does not only focus on the tidal component and that the surge component is being considered, it is just confusing as to how the analytical solution and numerical solution that includes surge should be compared. Response 1 does give a bit more clarity but I believe that the goal as quoted should be made much more clearly, even in the abstract as it is an important goal of the article.
Response 1: To make it clear, we enumerate the goals of the article in L37-53, p.2
- Validating the analytical model using the numerical model is a fine procedure although there can be arguments against it given that it does assume that the numerical representation is validated itself. I believe that you did this to some extent, although more validation would have been nice.
Response 2: Yes, we first validated the numerical model then validated the analytical model. I agree that more validations should be completed, however, at present the data-collecting work is still very difficult.
- Regarding the inverse problem approach what I am referring to is describing how you searched the parameter space. This space can be high-dimensional. If you simply evaluated the parameter space in a uniform, tensorial way then I would state that. The resolution of this evaluation though is important and I did not see that being mentioned. Also just saying that you are doing a least-squares fit would suffice if you would like.
Response 3: There are only two parameters and n. I think it is very simple and need not the tensorial way. We can only use enumeration method (not least-squares fit) by programme to obtain the best fit solution. The resolution was mentioned in the revised paper, it is 0.1 for and 0.001 for n And the range of the parameters can be determined through many classical hydrodynamic references. L7-12, p8
- The idea that analytical solutions, or I would say a "surrogate" or "reduced order model" can be extremely helpful is definitely something that is well accepted. I might suggest using some of this terminology to get a wider understanding of what it is you are doing given that this is effectively what you are doing.
Response : I like this terminology “surrogate” or "reduced order model", which was often used in the field of hydrologic science. I think this is the upgraded level of the model which we investigated at present. This idea is attracting and giving a direction for our model.
- Regarding response 4, I believe we agree but I would disagree still with the statement that "...numerical models do not provide direct insight...". Instead what I think you are saying is that one can extract data from a numerical model that can provide direct insight. In some sense your analytical model is building a regression model of the numerical solution and therefore is a direct result of the numerical model, which contains the insight, it just needs extraction from the resulting data.
Response 4: I change the saying as the reviewer’s suggestion in L12-16, p2
“The numerical results should be further investigated and extracted through the analytical solution to obtain deeper understanding on the effect by multiple controlling factors with the respect of tidal wave propagation.”
- Regarding response 6, I still believe you are being harsh but feel free to be careful. I know that many in the storm surge modeling community would be happy with this result.
Response 6: It’s ok and I agree with you now.
- Regarding response 7, I am not sure that it makes sense to state that "the meteorological conditions" can change. I believe this is an assumed capability for any numerical model and only lead to confusion.
Response 7: To avoid confusion, we deleted this sentence in L 19-20, p8.
- Regarding response 9, I would just make it clear in the article that this is what you are referring to.
Response 9: We referred to the value of , which is the parameter reflecting the tidal amplification or damping. (L1-2, p12 )
- Regarding response 10 + 11, you have identified my primary concern with the goals as stated in the article. The goals of the paper seem to me a bit contradictory due to this. I agree that it is interesting to think about the analytical model as perhaps a means for deriving further insight but without the processes involved it's difficult to see how this makes sense.
Response 10: We have rewritten and enumerate the goals of the paper according to the review’s suggestion.
Finally,
Once again, thank the reviewers for their patient suggestions, we think their help can greatly improve out article and make it much more clearly. Also, we express our gratitude to the editors for their encouragement and kind work.
Best Regards !
All authors in Nanjing, China.
